# MRI-Based Assessment of Brain Tumor Hypoxia: Correlation with Histology

**DOI:** 10.3390/cancers16010138

**Published:** 2023-12-27

**Authors:** Fatemeh Arzanforoosh, Maaike Van der Velden, Avery J. L. Berman, Sebastian R. Van der Voort, Eelke M. Bos, Joost W. Schouten, Arnaud J. P. E. Vincent, Johan M. Kros, Marion Smits, Esther A. H. Warnert

**Affiliations:** 1Department of Radiology & Nuclear Medicine, Erasmus MC, 3015 GD Rotterdam, The Netherlands; 2Brain Tumour Center, Erasmus MC Cancer Institute, 3015 GD Rotterdam, The Netherlands; 3Department of Physics, Carleton University, Ottawa, ON K1S 5B6, Canada; 4Institute of Mental Health Research, Royal Ottawa Mental Health Centre, University of Ottawa, Ottawa, ON K1N 6N5, Canada; 5Department of Neurosurgery, Erasmus MC Cancer Institute, 3015 GD Rotterdam, The Netherlands; 6Department of Pathology, Erasmus Medical Center, 3015 GD Rotterdam, The Netherlands; 7Medical Delta, 2629 JH Delft, The Netherlands

**Keywords:** brain tumor, cerebral hypoxia, blood vessels, magnetic resonance imaging, histology

## Abstract

**Simple Summary:**

Brain cells require a continuous and adequate supply of oxygen for optimal functioning; however, this balance is disrupted in the presence of brain tumors. The rapid growth of these tumors exceeds the capacity of the existing blood vessels, leading to areas of hypoxia. This condition contributes to accelerated tumor growth and diminishes the effectiveness of treatments. Utilizing MRI to non-invasively map hypoxia in the brain enables doctors to tailor treatment plans more effectively and to understand the tumor’s level of aggressiveness. In this study, we investigated the efficacy of a new MRI method, streamlined quantitative blood-oxygen-level-dependent (sqBOLD) MRI, in mapping hypoxia across different brain tumor types. Moreover, we utilize MRI to examine the vascular features of the tumors, aiming to elucidate the dynamics of oxygen delivery. In addition, this study includes microscopic evaluations of tumor biopsies, providing valuable insights into the hypoxic environments within the tumors.

**Abstract:**

Cerebral hypoxia significantly impacts the progression of brain tumors and their resistance to radiotherapy. This study employed streamlined quantitative blood-oxygen-level-dependent (sqBOLD) MRI to assess the oxygen extraction fraction (OEF)—a measure of how much oxygen is being extracted from vessels, with higher OEF values indicating hypoxia. Simultaneously, we utilized vessel size imaging (VSI) to evaluate microvascular dimensions and blood volume. A cohort of ten patients, divided between those with glioma and those with brain metastases, underwent a 3 Tesla MRI scan. We generated OEF, cerebral blood volume (CBV), and vessel size maps, which guided 3–4 targeted biopsies per patient. Subsequent histological analyses of these biopsies used hypoxia-inducible factor 1-alpha (HIF-1α) for hypoxia and CD31 for microvasculature assessment, followed by a correlation analysis between MRI and histological data. The results showed that while the sqBOLD model was generally applicable to brain tumors, it demonstrated discrepancies in some metastatic tumors, highlighting the need for model adjustments in these cases. The OEF, CBV, and vessel size maps provided insights into the tumor’s hypoxic condition, showing intertumoral and intratumoral heterogeneity. A significant relationship between MRI-derived measurements and histological data was only evident in the vessel size measurements (r = 0.68, *p* < 0.001).

## 1. Introduction

Maintaining a continuous and adequate supply of oxygen is crucial for the proper function of brain cells. Yet, in the context of brain tumors, this delicate equilibrium is frequently disrupted due to the rapid and uncontrolled proliferation of tumor cells, creating a demand for oxygen that surpasses the delivery capacity of the existing vascular network [1]. The lack of sufficient oxygen in the tumor microenvironment influences tumor development, encouraging angiogenesis, and facilitating the migration and invasion of tumor cells into adjacent healthy brain tissue. Importantly, cerebral hypoxia has been strongly correlated with resistance to treatment and is also associated with diminished patient survival rates [2].

Conventionally, positron emission tomography (PET) has been used to identify regions of hypoxia by utilizing specialized radiotracers that selectively bind and are retained in hypoxic tissue, thereby enabling their visualization [3]. However, the inherent need for radioactive tracers makes PET not only expensive but also less accessible for routine hypoxia imaging. In contrast, magnetic resonance imaging (MRI) techniques, devoid of radioactivity, have garnered increasing interest. Within MRI, oxygenation is determined by calculating the oxygen extraction fraction (OEF)—the proportion of oxygen utilized by the brain tissue to the total oxygen supplied through blood flow [4]. Among various methods, the quantitative blood-oxygenation-level-dependent (qBOLD) technique quantifies OEF by measuring the reversible transverse relaxation rate (R_2_′) and deoxygenated blood volume (DBV). R_2_′ indicates signal decay due to the paramagnetic characteristics of deoxyhemoglobin in the extravascular space [5]. 

Numerous studies have used the gradient echo sampling of spin echo (GESSE) sequence, often referred to as multi-parametric BOLD (mpBOLD), for OEF measurement. This approach measures both the irreversible transverse relaxation rate (R_2_) and transverse relaxation rate (R_2_*), to derive R_2_′ using the formula R_2_-R_2_* [6]. This approach has shown encouraging results for OEF determinations in glioma patients [1,7,8]. Alternatively, a newly developed streamlined quantitative BOLD (sqBOLD) employs the asymmetric spin echo (ASE) sequence with a fluid-attenuated inversion recovery (FLAIR) component (FLAIR-ASE) [9]. This not only allows for the direct measurement of R_2_′ but also efficiently suppresses the impact from CSF compartments. sqBOLD has been successful in measuring OEF in healthy volunteers and stroke patients [9,10] and we have recently demonstrated its preliminary application to measure OEF in gliomas [11]. In this study, our aim was to investigate the clinical applicability of the sqBOLD technique for measuring OEF across a wider spectrum of brain tumor types and delineate the conditions under which the model is effective or limited in its applicability. 

We further characterized the tumor environment by incorporating measurements of the tumor microvasculature using vessel size imaging (VSI), a contrast-agent-based acquisition technique that offers voxel-wise quantification of both average vessel size and cerebral blood volume CBV. VSI provides additional insight into the tumor tissue’s oxygenation capacity since suboptimal vascular morphology, such as the presence of disproportionately large microvasculature, leads to decreased efficient extraction of oxygen into tissue [12]. Moreover, the local utilization of oxygen is profoundly reliant on the volume of blood reaching the tissue, which can be assessed through CBV measurements [1,7,8].

Furthermore, to correlate our macroscopic MRI measurements with the microscopic tissue status, we conducted histopathological analyses on selected brain tumor biopsies. The analysis was particularly centered on the expression of hypoxia-inducible factor 1-alpha (HIF-1α), a transcription factor regulating gene expression in hypoxic conditions, and the expression of CD31 (also recognized as platelet endothelial cell adhesion molecule or PECAM-1). CD31 is predominantly expressed in endothelial cells and is widely used as a marker to stain blood vessel walls, making it crucial for vessel detection in various tissue types [13,14]. Lastly, we investigated the relationship between OEF and HIF-1α expression, given that both these factors are expected to be elevated in hypoxic regions. We also correlated vessel parameters, as assessed through histological evaluation of CD31 expression, with their corresponding MRI parameters.

## 2. Materials and Methods

### 2.1. Patients 

This study received approval from the local ethics board and was conducted in accordance with the Declaration of Helsinki. All patients provided written informed consent to participate. This was a prospective study, with patient enrollment taking place from August 2021 to October 2022, and with anonymized data collected from electronic medical records and tumor tissue samples obtained during surgical procedures.

The inclusion criteria for the study required participants to be aged 18 or above, with a pre-existing MRI diagnosis of glioma or brain metastasis, and referred for surgery. The diagnoses were subsequently confirmed through pathological examination. From a total of ten patients, aged between 40 and 78 years, five patients were diagnosed with glioma (mean age: 64 years, 3 males) and five patients with brain metastasis (mean age: 65 years, 2 males). Each participant underwent an MRI protocol, with the MRI scans being immediately analyzed to identify regions of interest within the tumor area for biopsy. Details of this process will be elaborated on in the following sections. Following the MRI scans, the patients underwent resection surgery the next day. During the surgery, an average of three targeted biopsy specimens were taken from predefined tumor locations and biopsies were forwarded to the pathology laboratory, where the hypoxia levels and microvascular dimensions of each biopsy were evaluated.

### 2.2. MR Data Acquisition

The MRI scans were conducted using a 3 Tesla (Discovery MR750, GE, Waukesha, WI, USA) whole-body scanner, equipped with a 32-channel head coil. The protocol incorporated both structural and physiological scans.

The structural scans included precontrast T1-weighted (T1W), T2-weighted (T2W), fluid-attenuated inversion recovery (T2W-FLAIR), and postcontrast T1W scans, all of which provided a whole-brain field of view (FOV). The parameters for both pre-and postcontrast T1W scans were identical (TE/TR: 2.1/6.1 ms; TI: 450 ms; voxel size: 1.0 × 1.0 × 1.0 mm^3^), with the postcontrast T1W scan conducted following the administration of 7.5 mmol of Gadovist, a gadolinium-based contrast agent (Bayer, Leverkusen, Germany). The T2W scan was performed with the parameters set to a TE/TR of 107/10,000 ms and a voxel size of 0.5 × 0.5 × 3.2 mm^3^. The T2W-FLAIR scan utilized the parameters of a TE/TR of 106/6000 ms, a TI of 1890 ms, and a voxel size of 0.6 × 0.5 × 0.5 mm^3^. In addition, the protocol included a diffusion-weighted imaging (DWI) scan to compute the apparent diffusion coefficient (ADC) (TE/TR: 63/5000 ms; voxel size: 1.0 × 1.0 × 3.0 mm^3^; 3 b-values: 0, 10, and 1000 s/mm^2^; FOV: whole brain). 

In the sqBOLD technique, MRI data were gathered using a FLAIR-ASE sequence with an EPI readout [15]. The FLAIR preparation served to minimize R_2_′ contamination from CSF [9]. Within the ASE sequence—a variant of the conventional spin echo sequence—the refocusing pulse is not centrally positioned between the excitation pulse and the subsequent readout (Figure 1a). Instead, it is shifted towards the excitation pulse by an interval of τ/2, causing the spin echo to be displaced from the echo time by τ. The shift values, τ, were: 0, 16, 20, 24, 28, 32, 36, 40, 44, 48, 52, 56, and 60 ms. The acquisition parameters were set to TE/TR, 74/8000 ms; TI, 2000 ms; voxel size, 2.3 × 2.3 × 3.0 mm^3^; slice thickness, 2.0 mm + 1.0 mm gap; and a total of 28 slices. The FLAIR-ASE scan was performed prior to the administration of the contrast agent and spanned 5 min and 12 s.

For the VSI technique, we performed dynamic susceptibility contrast imaging using a hybrid echo planar imaging (HEPI) sequence [16]. This MRI sequence simultaneously acquires both T2*-weighted images with gradient echo (HEPI-GRE) and T2-weighted images with spin echo (HEPI-SE). The acquisition parameters were set to TR, 1500 ms; TE(GRE)/TE(SE), 18.6/69 ms; voxel size, 1.88 × 1.88 × 4.00 mm^3^; slice thickness, 3.00 mm + 1.0 mm gap; a total of 28 slices; and a total sequence repetition of 122 [17]. The HEPI was performed during the administration of 7.5 mmol of gadolinium-based contrast agent (Gadovist, Bayer, Leverkusen, Germany), with the process commencing simultaneously with HEPI acquisition. The duration of this scan was 3 min and 3 s.

### 2.3. MR Data Analysis

The brain volumes collected using the FLAIR-ASE sequence first underwent motion correction. All volumes with varying τ values were temporally concatenated and then aligned to the first volume of the FLAIR-ASE sequence (τ = 0) using FSL’s MCFLIRT rigid registration tool, resulting in a 4D MRI dataset [18]. From this dataset, we measured and logarithmically transformed the MRI signal of each voxel. This transformed signal was then plotted against the associated shift (t) values from which the MRI volumes were derived. We named this transformed signal the ‘FLAIR-ASE signal’ and will use this term throughout the study. For this FLAIR-ASE signal to align with the sqBOLD model (Figure 1b), the peak of the FLAIR-ASE signal should occur at τ = 0. For τ > 0, the FLAIR-ASE signal is characterized by two distinct patterns, namely a quadratic regime (for τ < τ_c_) and a linear regime (for τ > τ_c_). τ_c_ is the transition time between these regimes, with a conservative estimate of τ_c_ = 16 ms in gray matter [19]. The FLAIR-ASE signal is described in Equation (1):(1)$$Ln(S\left(\tau \right))=\left\{\begin{array}{l} {ln(S}_{0})-\left(R_{2}^{}\cdot TE\right)-\left(0.3\cdot \frac{{\left(R_{2}^{\prime }\cdot \tau \right)}^{2}}{DBV}\right)\;\;\;\;\;\;\;\;\;\;\;\;\tau <\tau_{c}\\ {ln(S}_{0})-\left(R_{2}^{}\cdot TE\right)+(DBV-R_{2}^{\prime }\cdot \tau )\;\;\;\;\;\;\;\;\;\;\tau >\tau_{c} \end{array}\right.$$
where the factor S_0_ accounts for proton density, coil sensitivity, and other factors, R_2_ is the irreversible transverse relaxation rate, and TE is the echo time. The FLAIR-ASE data were fit to Equation (1) to estimate R_2_′ and DBV using the Quantiphyse toolbox, based on Bayesian model fitting [20]. Once R_2_′ and DBV were obtained, OEF was calculated using Equation (2): (2)$$OEF=\frac{{3R}_{2}^{\prime }}{4\pi \cdot \gamma B_{0}\cdot {∆\chi }_{0}\cdot Hct\cdot DBV}$$
where γ is the proton gyromagnetic ratio of 267.5 × 10^6^ rad·T^–1^·s^–1^, B_0_ is the field strength (3T in this study), ${∆\chi }_{0}$ is the susceptibility difference between fully oxygenated and fully deoxygenated red blood cells (0.264 ppm in cgs units), and Hct is blood hematocrit, assumed to be 0.4 [21]. 

Finally, the R_2_′, DBV, and OEF maps were transformed to the postcontrast T1W image space using linear registration with Elastix.

The preprocessing of both HEPI-SE and HEPI-GRE data included discarding the first four brain volumes to ensure FLAIR-ASE signal stabilization. Motion correction for both HEPI-SE and HEPI-GRE was performed using MCFLIRT, wherein each series was rigidly aligned to the fifth volume of its respective sequence. In-house code developed in Python 3.6 (http://www.python.org, accessed on 4 April 2022) was used for acquiring vessel size and CBV maps from coregistered HEPI-GRE and HEPI-GRE datasets [17]. While this process is described comprehensively in our previous work, we provide a brief summary here. Signal–time curves were plotted for each voxel for both HEPI-GRE and HEPI-SE data separately. These curves were converted into transverse relaxation rates of ∆R_2_*(t) and ∆R_2_(t), respectively. The CBV map was estimated voxelwise from ∆R_2_*(t) by measuring the trapezoidal integration in the time interval of entrance time to exit time of the bolus of the contrast agent. Given that most of the tumors demonstrated a disrupted blood–brain barrier, evident by enhancement in the postcontrast T1W, the Boxerman–Schmainda–Weisskoff (BSW) leakage correction algorithm [22] was applied when analyzing CBV, as previously described [23]. Normalization of the CBV map was achieved by dividing it by the mean CBV value of the contralateral WM voxels. Hereafter in this paper, the term CBV will refer to the normalized CBV. The estimation of mean vessel size for each voxel was based on the Kiselev model [12] and represented by Equation (3): (3)$${Vessel\;Size}_{}=0.867{\left(CBV\times ADC\right)}^{1/2}\left[\frac{{∆R}_{2}^{*}}{{{(∆R}_{2}^{})}^{3/2}}\right]$$
where ∆R_2_*(t) and ∆R_2_(t) are transverse relaxation rates over time, representing their maximum values achieved over the observed time period. ADC is an apparent diffusion coefficient acquired from the diffusion-weighted scan. 

Finally, CBV and vessel size maps were linearly transformed to the postcontrast T1W image space using Elastix (version 4.8). 

### 2.4. Volume of Interest (VOI) Segmentation

For tumor segmentation, we employed an in-house developed algorithm, ‘Glioseg’ (described in our earlier work) [17]. This algorithm integrates multiple publicly available segmentation algorithms and is applied to the pre- and postcontrast T1W, T2W, and T2W-FLAIR images [24,25,26]. It generates binary masks of the tumor’s VOIs, which contain edema, nonenhancing, enhancing, and necrosis VOIs, depending on the type of tumor under examination. In this context, the edema and nonenhancing VOIs correspond to regions that do not exhibit contrast uptake but are hyperintense on T2W/T2W-FLAIR, referred to as the edema VOIs in glioblastoma and brain metastasis, and as the nonenhancing VOIs in low-grade glioma. The enhancing VOIs represent regions showing contrast uptake on postcontrast T1W images, and necrosis VOIs correspond to centrally located nonenhancing areas surrounded by the enhancing VOI.

To generate masks for the contralateral white matter (contra-WM) and gray matter (contra-GM), we used FMRIB’s automated segmentation tool (FAST, Version 6.0.5) on the precontrast T1W images after brain extraction with the HD-BET tool [27]. This resulted in tissue probability maps for gray matter, white matter, and CSF. Next, we selected the white matter and gray matter probability maps from the hemisphere contralateral to the tumor and binarized these using a probability threshold of 0.9 to minimize partial volume effects. Finally, all masks including edema, nonenhancing, enhancing, and necrosis VOIs as well as contra-GM and contra-WM VOIs were linearly registered to the postcontrast T1W images space using the registration tool of Elastix (version 4.8) [28]. 

### 2.5. VOI-Based MRI Data Evaluation

The FLAIR-ASE signals were averaged across different tumor VOIs—edema, nonenhancing, enhancing, and necrosis—for each patient. The averaged FLAIR-ASE signals from various VOIs were adjusted using additive or subtractive methods to align with a unified baseline for visualization. This baseline, for each patient, was determined by the initial point of the average FLAIR-ASE signal from contra-GM VOI. The aim here was to determine whether the averaged FLAIR-ASE signal in each VOI followed the expected decay pattern depicted in Figure 1b. The FLAIR-ASE data were analyzed using Bayesian model fitting through the Quantiphyse platform. The primary indication of the confidence of the fit in the variational Bayesian (VB) method is the free energy, which the algorithm aims to minimize. Should the Bayesian model fitting be unsuccessful in accurately matching the FLAIR-ASE signal to the sqBOLD model for specific voxels, these voxels are assigned a value of 1 in the error mask, thereby marking them as ‘failure voxels’.

All maps, encompassing R_2_′, DBV, OEF, CBV, and vessel size, underwent both qualitative and quantitative evaluations. In the qualitative evaluation, all of these maps were visually inspected by an MR physicist alongside an experienced radiologist. For the quantitative analysis, the median and interquartile range (IQR) of R_2_′, DBV, OEF, CBV, and vessel size were computed across the edema, nonenhancing, enhancing, and necrosis VOIs for each patient. Any ‘failure voxels’ highlighted within the error mask were excluded from this assessment.

### 2.6. Image-Guided Biopsy Procedure

We defined discrete clusters within the brain tumor for biopsy, with each cluster containing a minimum of nine voxels (arranged cubically) on postcontrast T1W images. These clusters are identified by parameters like OEF, CBV, and vessel size, with high values defined as above the 80th percentile and low values below the 20th percentile. The first step in our methodology was to delineate two sets of clusters based on OEF values: one representing high OEF and another representing low OEF. Subsequently, we followed a similar process for CBV values. When a CBV-derived cluster exhibited overlap with any of the previously determined OEF-derived clusters, we prioritized that overlapping combination. A similar method of identification and prioritization was applied for clusters based on vessel size. These designated clusters were then presented to the neurosurgeons, who selected specific clusters and their precise biopsy locations. Typically, 2 to 4 biopsy locations were chosen per patient. Tumors in sensitive or difficult-to-access areas had fewer biopsy points to minimize risk, whereas larger or more accessible tumors had multiple sites to capture the tumor’s heterogeneity. It is worth mentioning that, according to the protocol and as a precautionary measure, biopsies from the edema VOIs were prohibited.

The planned biopsy locations were identified on postcontrast T1W images and subsequently integrated into the Brainlab neuronavigation software (version 6.1.1.210, Brainlab AG, Munich, Germany). To facilitate surgical guidance, the surgeon delineated the trajectory for each intended biopsy location and saved these within the neuronavigation system. During the surgical procedure, post-craniotomy but prior to dural incision—to mitigate potential brain shift—the biopsy needle was accurately introduced according to the predetermined trajectory using the Brainlab VarioGuide system, and tissue samples were procured. The biopsy specimens, roughly cylindrical in shape with an approximate height of 1 cm and a diameter of 1 mm, were placed and labeled accordingly in tubes filled with a 10% buffered formalin solution and subsequently dispatched to the pathology laboratory for analysis.

### 2.7. Histological Analysis

Tumor biopsies were embedded in paraffin, sectioned into about five 4 µm thick slices, and mounted on slides. After deparaffinization and rehydration, one slice from each sample, averaging 14 mm^2^ in area, was stained with hematoxylin and eosin (H&E) for cellular morphology visualization. Adjacent slices underwent CD31 staining for endothelial cells and vascular patterns and HIF-1α staining for hypoxia indicators. Slides were scanned to produce high-resolution whole-slide images and subsequently analyzed using QuPath software (version 0.3.2) [29]. In CD31-stained images, in each identified vessel, the diameter was measured, and an average diameter was computed within hotspot ROIs. Vessel density was determined by dividing the total vessel area by the total tissue area within these hotspot ROIs. Quantification of HIF-1α staining employed a scoring system that took into account both staining intensity and the percentage of positively stained cells. To ensure uniformity in staining, all samples were processed in a single batch. Comprehensive details regarding the staining protocols and quantification methodologies are available in Appendix B.

### 2.8. MRI Measurements in the Target Voxels

To evaluate the correlation between MRI and histological findings, values for the MRI-derived parameters (OEF, CBV, and vessel size) were obtained by averaging these metrics from each target voxel from which biopsies were taken, and their adjacent nine voxels in three dimensions in postcontrast T1W images. This methodology was implemented to mitigate potential inaccuracies stemming from relying solely on single-voxel evaluations and to accommodate potential discrepancies caused by intraoperative brain shifts and MRI data registration challenges. 

### 2.9. Statistical Analysis

All statistical analyses were conducted using SciPy (version 1.10.1) software in Python (version 3.6). First, the Kruskal–Wallis H test, a non-parametric statistical test, was employed to evaluate the differences in various MRI biomarkers and histology parameters across three distinct regions of interest: nonenhancing, enhancing, and necrosis. Post hoc pairwise comparisons were conducted using the Mann–Whitney U test to further investigate specific differences between these regions, and the results were annotated with their corresponding *p*-values directly on the bar plots.

Furthermore, correlations between MRI-derived measurements and histological counterparts were evaluated using a non-parametric Spearman correlation test. Examined correlations included OEF vs. HIF-1α score, CBV vs. vessel density, and vessel size vs. vessel diameter. For each analysis, the correlation coefficient (r) and its associated *p*-value were recorded, with a significance threshold set at *p* < 0.05. 

## 3. Results

### 3.1. Patient and Biopsy Characteristics

In this study, we collected a total of 33 targeted biopsy samples from 10 patients, with ages ranging from 40 to 78 years. The cohort consisted of six males and four females, diagnosed with various brain pathologies: two with oligodendroglioma, one with astrocytoma, two with glioblastoma, and others with brain metastases including one from lung carcinoma, three from adenocarcinoma, and one from melanoma. Table 1 provides information on each patient’s age, sex, and specific diagnosis, as well as the distribution of biopsy samples across necrotic VOIs (*n* = 8), enhancing VOIs (*n* = 14), and nonenhancing VOIs of the tumors (*n* = 11). 

### 3.2. Characteristics of the FLAIR-ASE Signal

In Figure 2, the mean FLAIR-ASE signal is depicted across the segmented tumor VOIs—edema, nonenhancing, enhancing, and necrosis—in five exemplary patients: oligodendroglioma, astrocytoma, two brain metastases, and glioblastoma. Appendix A provides the mean FLAIR-ASE signal for each VOI for other patients. To facilitate comparison among VOIs, the average FLAIR-ASE signal from voxels in the contra-GM is provided as a reference and the tumor FLAIR-ASE signals were normalized so that they started at the same value as the contra-GM FLAIR-ASE signal, τ = 0. 

In glioma, specific observations were made based on their subtypes. In the case of oligodendroglioma (patient 1 and patient 6), the nonenhancing VOI aligned with the predicted sqBOLD model. The decay rate in these nonenhancing VOIs was marginally lower than that observed in the contra-GM VOI. However, in the small enhancing VOI of patient 6, the decay mirrored that of the contra-GM VOI. For astrocytoma (patient 2), the average FLAIR-ASE signal within the nonenhancing VOIs shows a decay so minimal that it could be regarded as non-existent, especially when contrasted with the decay in contra-GM voxels. In glioblastoma (patients 8 and 10), where three distinct VOIs—edema, enhancing, and necrosis—were identified, the edema VOIs consistently showed slow FLAIR-ASE signal decay, contrasting sharply with the rapid decay in patient 10’s enhancing VOI and patient 8’s enhancing and necrosis VOI. 

In our analysis of brain metastasis across five patients, the observed FLAIR-ASE signal patterns could be categorized into two distinct groups. The first category, encompassing patients 3, 5, and 7, exhibited FLAIR-ASE signals in line with the sqBOLD model. Here, the nonenhancing or edema VOIs showcased minimal decay, contrasting with the more pronounced decay in the enhancing and necrosis VOIs. The second group, comprising patients 4 and 9, demonstrated unique FLAIR-ASE signal behaviors. Specifically, for patient 4, the contra-GM FLAIR-ASE signal conformed to the sqBOLD model, but the FLAIR-ASE signals from the edema, enhancing, and necrosis VOIs diverged, peaking notably at (τ = 28 ms), as depicted in Figure 2. In the case of patient 9, both the necrosis and enhancing VOIs demonstrated the expected FLAIR-ASE signal decay. However, the edema FLAIR-ASE signal remained relatively constant, with a noticeable spike at (τ = 20 ms).

Voxels in which the FLAIR-ASE signal did not fit the expected sqBOLD decay are displayed in Figure 3 as error masks superimposed on the postcontrast T1W images. The voxels with failed fits are dispersed somewhat sporadically throughout the brain, with a notable concentration within the CSF voxels. Notably, error voxels accumulated in substantial portions of edema VOIs related to metastasis in patient 4 and patient 9. This observation is consistent with the results in Figure 2 for patients 4 and patient 9, where the average FLAIR-ASE signals in the relevant VOIs deviated from the expected sqBOLD model. The tumor area in patient 2 (astrocytoma) had a notable number of error voxels, as the FLAIR-ASE signal stayed relatively stable without showing the anticipated decay.

### 3.3. Features of MRI Parametric Maps

The sqBOLD maps were visually inspected by an experienced radiologist and an MRI physicist. Figure 4 highlights one slice of sqBOLD-derived maps of R_2_′, DBV, and OEF, complemented by perfusion maps of CBV and vessel size for five selected patients, consistent with those featured in Figure 2. The final row in this figure magnifies the tumor region from one patient for enhanced clarity. A similar presentation of slices for the remaining patients can be found in Appendix A. Voxels identified in the error masks have been zeroed out in these maps, which is particularly evident in the edema VOIs of patients 4 and 9, both diagnosed with brain metastasis. 

As seen in Figure 4, R_2_′, DBV, and OEF maps exhibit contrasts between the normal brain tissue and tumor regions. While R_2_′ and OEF maps present enhanced contrast and clarity, the contrast in the DBV map appears comparatively suboptimal. Likewise, the CBV and vessel size maps both present contrasts that are relatively similar to each other. However, they do not always correspond with the contrasts seen in the R_2_′ and OEF maps. For instance, in the case of patient 9 (brain metastasis), all maps—namely R_2_′, DBV, and OEF as well as CBV and vessel size—emphasize the contrast within the enhancing and necrosis VOIs in comparison to the rest of the brain. Similarly for patient 10 diagnosed with glioblastoma, the R_2_′, DBV, and OEF maps show pronounced brightness in the enhancing VOIs, which also extends somewhat into the necrotic VOIs. However, the CBV and vessel size maps predominantly display this brightness in the enhancing VOIs, with the necrotic VOIs showing no such effect.

In Table 2, we provided the mean values and standard deviations across patients for R_2_′, DBV, OEF, CBV, and vessel size in contra-GM and various tumor VOIs. For a detailed breakdown of each patient’s data, which include median values and the interquartile ranges for their respective VOIs, please see Appendix A. For most patients, the R_2_′, DBV, and OEF measurements in the contra-GM VOIs consistently fell within a moderate range. Nonetheless, there were standout observations: Patients 8 and 10, both diagnosed with glioblastoma, exhibited notably high values for R_2_′, DBV, and OEF within the contra-GM VOI. Conversely, patient 3 (brain metastasis) recorded the lowest value. The CBV and vessel size measurements in the contra-GM maintained a steady average, reflecting their stability across the cohort. Turning our attention to the edema and nonenhancing VOIs, it was evident that they predominantly registered low values across all five maps. Their narrow standard deviation signals a consistency in these results among the patient group. In contrast, the enhancing and necrosis VOIs reported higher average values and variability (as indicated by the standard deviation) for R_2_′, DBV, and OEF. This variation underscores the individual characteristics present in each patient’s condition. When evaluating CBV and vessel size, we discerned elevated metrics in the nonenhancing VOIs, a trend that was not mirrored in the necrosis VOIs.

### 3.4. Histological Findings

All patient biopsies were successfully stained with H&E, CD31, and HIF-1α. Figure 5 presents three randomly selected biopsies from each distinct VOI (nonenhancing, enhancing, and necrosis VOIs). Importantly, these samples were selected from regions outside those identified in the error masks. The H&E staining depicts cell nuclei in varying shades from blue to dark blue and colors the cytoplasm and extracellular matrix in tones ranging from pink to red. For the CD31 and HIF-1α stains, the blue represents cell nuclei. The brown shade in the CD31-stained samples identifies the presence of endothelial cells, signifying blood vessels. Meanwhile, in the HIF-1α-stained samples, regions marked brown indicate active HIF-1α, signifying a response to hypoxic conditions.

The first row in Figure 5 displays samples from a nonenhancing VOI. There is low nuclear density (evident from H&E), reduced vessel density (from CD31), and a scant presence of HIF-1α. The middle row showcases samples from an enhancing VOI, revealing high nuclear density (as per H&E), abundant blood vessels (from CD31), and extensive HIF-1α expression. The final row features a biopsy from a necrosis VOI, where the staining patterns for H&E, CD31, and HIF-1α appear patchy. The H&E-stained section exhibits two distinct areas. The first one is populated with eosinophilic debris lacking a discernible cellular structure, observed in the upper third. This particular region is mirrored in the CD31 staining, showing a high vessel density, often with a thrombotic character, and no HIF-1α expression. In contrast, the neighboring area displays dense nuclei (indicative of extensive cell proliferation or cell migration) coupled with moderate vessel density and a high HIF-1α expression level observed in the lower third. 

### 3.5. Correlation between MRI and Histology 

Assessment of MRI-derived parameters (OEF, CBV, and vessel size) in conjunction with their histological analogs (HIF-1α score, vessel density, and vessel diameter) yielded the following significant findings (Figure 6): Differences in OEF were evident between the nonenhancing and enhancing VOIs (*p* = 0.02, Mann–Whitney U test) and between the nonenhancing and necrosis VOIs (*p* < 0.001, Mann–Whitney U test). Conversely, HIF-1α scores did not demonstrate any significant variances across VOIs. The CBV value in the enhancing VOIs was markedly higher than in the nonenhancing VOIs (*p* = 0.02), whereas no significant variations were observed in vessel density across the VOIs. Both vessel size and vessel diameter exhibited significant differences between the nonenhancing and enhancing VOIs (*p* < 0.001, Mann–Whitney U test), with vessel diameter further showing a distinct difference between the nonenhancing and necrosis areas (*p* < 0.001, Mann–Whitney U test).

Figure 7 examines the direct correlation between the MRI and histological measurements. No significant voxel-wise correlation was found between MRI measurements of the OEF and the HIF-1α scores in biopsy samples (r = 0.15, *p* = 0.40, Spearman correlation test), nor between CBV and vessel density in histology (r = 0.26, *p* = 0.15, Spearman correlation test). However, a robust positive correlation was established between MRI measurements of vessel size and histology-derived measurements of vessel diameter (r = 0.67, *p* < 0.001, Spearman correlation test).

## 4. Discussion

The present study introduces, for the first time, the application of the sqBOLD technique using the FLAIR-ASE MRI sequence to measure oxygen extraction fraction (OEF) in brain tumor patients. In the initial stages of our study, we evaluated the FLAIR-ASE signal to see if it matched the expected pattern, finding that signal decay generally aligned with the model in most tumor regions [9]. However, exceptions were noted in some cases of brain metastases, the details of which will be elaborated on below. The R_2_′, DBV, and OEF were measured exclusively in areas showing the expected signal decay, and in these areas, our findings closely aligned with previous research that observed elevated contrast in the enhancing and certain necrotic areas of gliomas. Conversely, diminished contrast was observed in the tumor edematous and nonenhancing regions [7,13,30], findings that mirror our own observations. As illustrated in Figure 4, the resulting R_2_′, DBV, and OEF maps exhibit a strong interrelation. A rise in OEF indicates increased oxygen extraction from the blood, leading to heightened deoxyhemoglobin concentrations, which directly influence both DBV and R_2_′ measurements. Therefore, for the remainder of the discussion, we will focus on OEF in different areas of the tumors.

In edema VOIs, we consistently observed low values of OEF, CBV, and vessel size across all tumor types (see Table 2). This is consistent with expectations, as fluid-filled tissues metabolize oxygen more slowly, leading to a reduction in OEF [31]. However, it is essential to highlight that the sqBOLD model assumes a static dephasing regime and negates water diffusion [32]. Such assumptions may not hold in edematous regions because of the excess fluid accumulation in this area. Additionally, the decreased CBV and vessel size in the edema VOIs might suggest the influence of extracellular fluid accumulation, which could exert pressure on adjacent vessels, potentially impacting blood flow [33]. Owing to safety concerns, biopsies were not performed in the edema regions, thereby limiting our ability to conduct a detailed microscopic examination.

The increased OEF, vessel size, and CBV in enhancing VOIs can be attributed to irregular and rapid cellular proliferation, necessitating higher oxygen and nutrient demands. This, in turn, stimulates enhanced oxygen extraction and the process of angiogenesis [34]. Histological examinations showed accelerated cellular growth, increased vascular density, and enlarged vessel diameter in some samples within the enhancing VOIs (as seen in Figure 5). Still, it is essential to note that not all samples align with these observations. The variability may be attributed to factors, such as a tumor’s genetic makeup, that influence the tumor’s biological behavior and histological appearance [35].

Heterogeneous elevated mean OEF in necrosis VOIs was observed across patients, contrasted with decreased CBV and vessel sizes (Table 2). Histological examinations of the necrotic regions confirmed characteristics such as a pronounced absence of nuclei and vascular abnormalities. These abnormalities included thickened basement membranes and endothelial cell accumulation in the vessel, both of which can impede efficient blood flow. This is consistent with the decreased CBV and vessel dimensions observed on MRI, as illustrated in the upper section of Figure 5 within the necrotic VOI. Adjacent to these regions, viable tumor cells are typically present, exhibiting significant HIF-1α expression, as depicted in the lower section of Figure 5 within the necrotic VOI. The relatively high OEF values in necrosis are thus likely due to an increase in oxygen consumption by the remaining viable cells, combined with potential reductions or stagnation in blood flow, resulting in a significant accumulation of deoxygenated blood [31,36].

### 4.1. Correlation between MRI and Histology

The absence of a significant correlation between OEF and HIF-1α seems counterintuitive since elevations in both parameters typically signify hypoxia. However, it is important to note that they each capture different facets of the hypoxic condition. HIF-1α is a transcription factor upregulated in response to reduced oxygen concentrations, while OEF quantifies the fraction of oxygen that tissues absorb from the bloodstream. These two metrics might not always vary in tandem. For example, during severe or sustained hypoxia, impaired oxygen extraction mechanisms can lead to decreased OEF levels, despite active upregulation of HIF-1α [8]. To the best of our knowledge, no study has reported a direct positive correlation between OEF and HIF-1α expression in brain tumors. This includes the study by Tóth et al., which, while investigating OEF and HIF-1α expression, did not report a direct relationship between them [13]. Similarly, Yao et al. established a correlation between R_2_′ measurements, related to OEF by Equation (2), and HIF-1α expression [37], but only in high-grade gliomas and not in low-grade ones, both of which were considered in the current study. However, while we did not observe a direct one-to-one correlation between OEF and HIF-1α when analyzing the data by regions, we identified a consistent trend in both OEF and HIF-1α, across different VOIs.

In addition, this study found a positive correlation between vessel size from MRI data and the diameter of tissue sample vessels, as shown in both region-based (Figure 6) and individual sample data (Figure 7), consistent with other studies [14]. Yet, no correlation emerged between CBV (an MRI metric for vessel density) and density derived from CD31-stained samples. This may stem from CBV focusing on blood volume in vessels, while CD31 staining emphasizes vessel structure without indicating blood flow. Many studies show varied results linking vessel density from CD31 staining and CBV [38,39], though comparing our findings with those of previous studies is complicated due to the diverse tissue types and tumor inclusions in our research.

Another point of consideration here is the challenge inherent in correlating MRI data with histopathological findings. This stems from the spatial resolution disparities between MRI and histopathological sections. Specifically, an MRI voxel represents a 3D space on the order of mm^3^, providing a localized, weighted average of the signal originating from tissue across several mm^3^. In contrast, histology captures detailed information on a 2D slice of tissue only a few micrometers thick. This discrepancy in scale, combined with the tissue heterogeneity, means that the specific details observed in histopathological sections may not fully represent the broader signals captured by MRI. This could be a contributing factor to the difficulty in achieving a one-to-one correlation between MRI and histological findings.

### 4.2. Limitations

During the course of this study, we encountered several challenges that require further discussion. A specific issue was highlighted by the data from patient P4, diagnosed with a metastasis tumor. Typically, the peak of the FLAIR-ASE signal is expected at a shift of zero. This was consistent in the contralateral GM voxels for P4; however, within tumor regions, this peak deviated from τ = 0. A potential explanation for this deviation may be linked to the inherent assumptions of the qBOLD model, especially its reliance on the static dephasing regime which overlooks diffusion effects. Given that this patient exhibited a pronounced edematous area, known to be fluid-rich, this might account for the atypical FLAIR-ASE signal behavior within the tumor region. These variations indicate that the interpretation of FLAIR-ASE signals and OEF measurements may need to be tailored to individual patients rather than adhering to a universal model. Incorporating metrics like the ADC, which assesses water diffusion in tissues, enables us to ascertain the degree to which the qBOLD model’s assumptions are violated. It is worth mentioning that the disparities in how various tumor types follow the sqBOLD model could provide valuable diagnostic cues, potentially aiding in distinguishing between different brain tumor classifications.

The other issue we faced in this study is that the OEF values, obtained using the sqBOLD technique, frequently exceeded the traditional 0–100% range. These abnormally high values were predominantly noted in areas of enhancement, at times in necrosis regions, in CSF voxels, and in regions marked by air–tissue interfaces, as exemplified in the results from patient 8 (Appendix A). Within the qBOLD methodology, the prevailing assumption is that the derived values are solely indicative of deoxygenated blood. In contrast to this assumption, these parameters also exhibit sensitivity to macroscopic B0 inhomogeneities. To address this, corrective measures are essential, either prospectively—using the z-shimming technique [9]—or retrospectively, with a high-resolution field map [40]. Nevertheless, it is essential to note that even without these corrections, the derived OEF map remains informative, producing contrasts across various tumor regions consistent with those reported in other studies. To ensure more accurate quantitative OEF maps in the future, it is crucial to integrate these corrections.

## 5. Conclusions

In our study, we aimed to measure OEF across a diverse set of brain tumors, spanning both low- and high-grade gliomas, as well as brain metastases, using the sqBOLD technique, and to compare the MRI findings to histology from the same patients. Our observations indicated that, although the sqBOLD technique was applicable to a majority of the tumors assessed, it necessitates further optimization to ensure consistent results across all tumor types; yet the disparities in how various tumor types resonate with the sqBOLD model might help distinguish brain tumor types. Given its non-invasive and quantitative attributes, sqBOLD emerges as a promising tool for clinical application, facilitating ongoing patient surveillance and the evaluation of therapeutic interventions. Future research would benefit from engaging more expansive and consistent patient cohorts to reinforce the robustness and broad applicability of our conclusions.

## Figures and Tables

**Figure 1 cancers-16-00138-f001:**
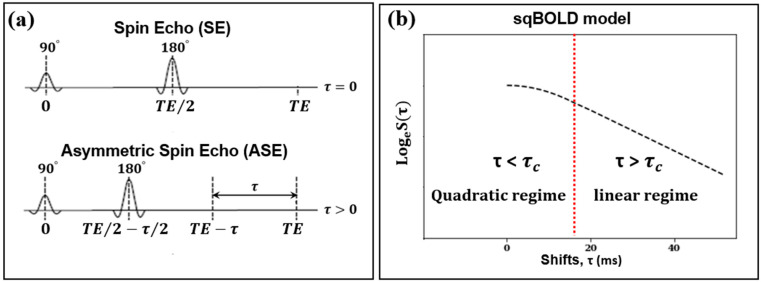
(**a**) The left diagram illustrates the spin echo (SE) sequence with a 180° refocusing pulse occurring at half the echo time (TE), and the asymmetric spin echo (ASE) sequence where the refocusing pulse is shifted earlier by a time τ/2, while the TE remains unchanged. (**b**) The right diagram represents the streamlined quantitative BOLD signal decay (in logarithm) as a function of the ASE shift (τ). The signal peaks when the shift is zero. The signal transitions from a quadratic regime to a linear regime at a shift of 16 ms, marked with $\tau_{c}$, highlighted by the dashed red line.

**Figure 2 cancers-16-00138-f002:**
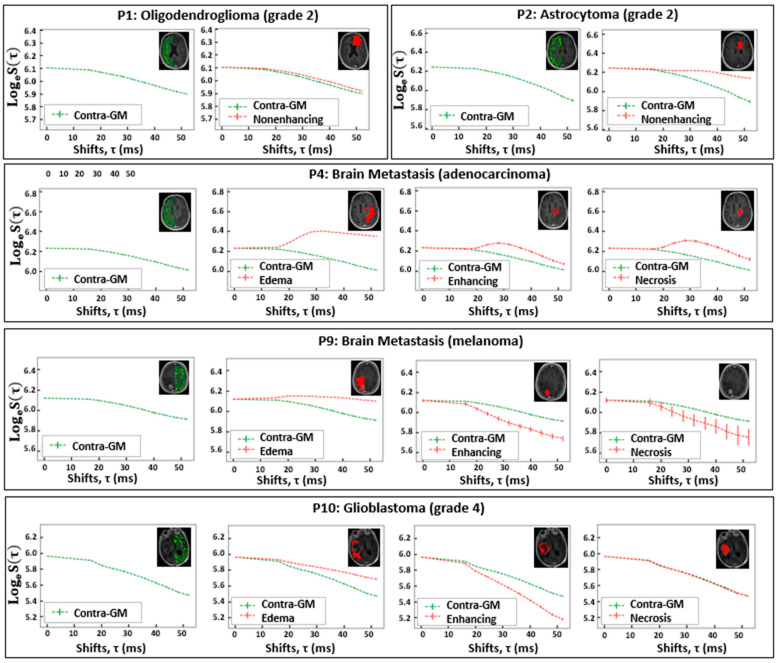
The VOI-averaged FLAIR-ASE signal across various VOIs for five patients with brain tumors. The VOIs are contra-GM, nonenhancing, edema, enhancing, and necrosis. However, not all VOIs are relevant to each tumor type. Error bars represent ± the standard deviation of the signal over the VOI. VOI: volume of interest; contra-GM: contralateral gray matter; FLAIR-ASE: Fluid-attenuated inversion recovery–asymmetric spin echo.

**Figure 3 cancers-16-00138-f003:**
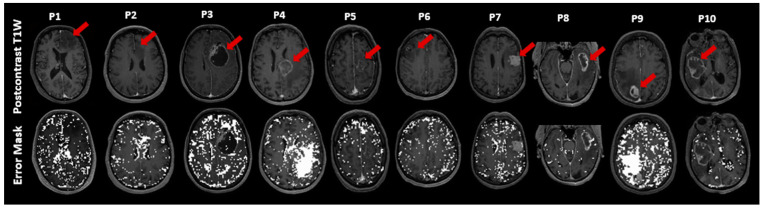
This figure illustrates exemplary slices of postcontrast T1-weighted (T1W) images for ten patients (**P1**–**P10**). For each patient, the T1W image is placed above with its corresponding error mask overlaid on the postcontrast T1W below, highlighting ‘failure voxels’, which are voxels that failed to fit the sqBOLD model. The tumor’s location in each slice is marked with a red arrow. sqBOLD: streamlined quantitative blood-oxygenation-level-dependent.

**Figure 4 cancers-16-00138-f004:**
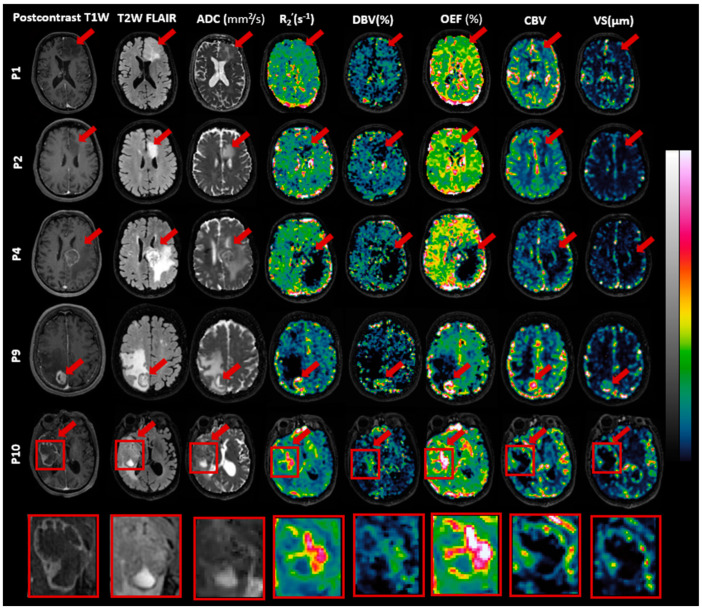
Exemplary slices of images including postcontrast T1-weighted (T1W), T2-weighted fluid-attenuated inversion recovery (T2W FLAIR), apparent diffusion coefficient (ADC), reversible transverse relaxation rate (R_2_′), deoxygenated blood volume (DBV), oxygen extraction fraction (OEF), cerebral blood volume (CBV), and vessel size are presented for patients P1 (oligodendroglioma), P2 (astrocytoma), P4 and P9 (brain metastasis), and P10 (glioblastoma). Red arrows highlight the tumors. Intensity scales vary across images, with ADC ranging from 0 to 3 × 10^−3^ mm^2^/s, R_2_′ from 0 to 20 s^−1^ (0–40 s^−1^ for P10), DBV from 0 to 50% (0–100% for P10), OEF from 0 to 100% (0–200% for P10), CBV from 0 to 10, and vessel size from 0 to 300 µm. A detailed view of P10’s tumor area is included.

**Figure 5 cancers-16-00138-f005:**
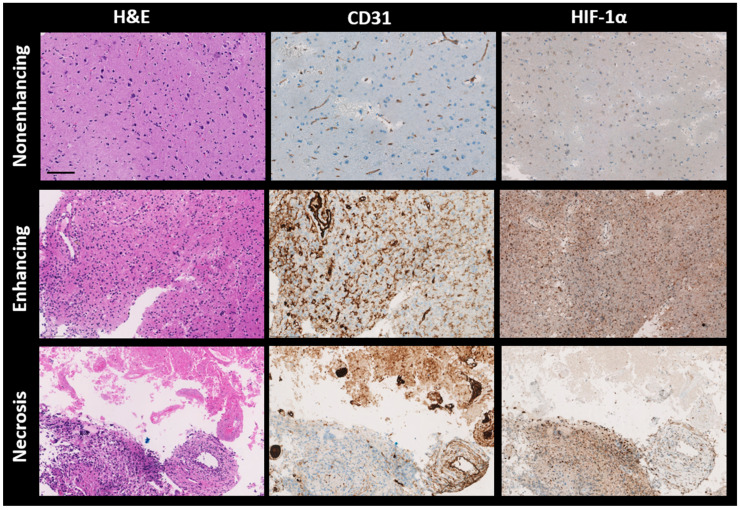
Representative histological sections of brain tumor tissues from different patients, stained with hematoxylin and eosin (H&E), CD31, and HIF-1α antibodies, are shown from left to right. Top to bottom, the sections are from the nonenhancing, enhancing, and necrotic volumes of interest (VOIs) of brain metastasis (P5), glioblastoma (P10), and glioblastoma (P8), respectively. H&E staining colors nuclei dark blue and cytoplasm pink, while both CD31 and HIF-1α stain nuclei light blue. CD31 specifically highlights endothelial cells in brown, and HIF-1α similarly marks hypoxia-response elements in brown. H&E—hematoxylin and eosin; CD31—cluster of differentiation 31; HIF-1α—hypoxia-inducible factor 1-alpha, Scale bar: 100 μm.

**Figure 6 cancers-16-00138-f006:**
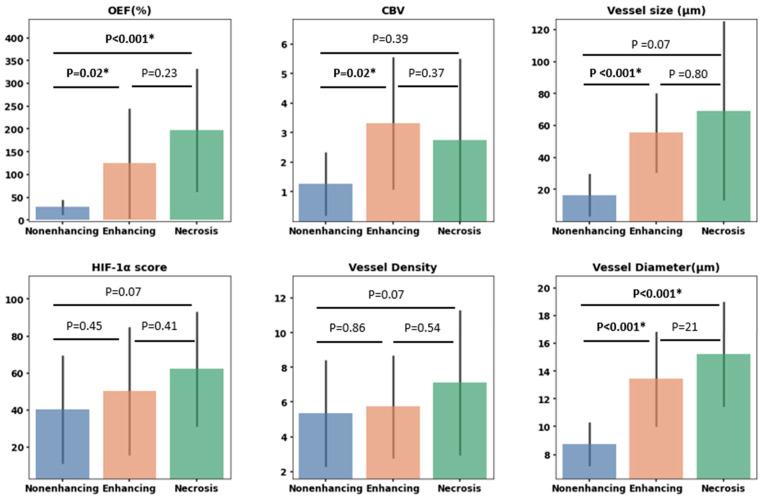
Bar plots illustrating measurements for various parameters. The top row displays oxygen extraction fraction (OEF), cerebral blood volume (CBV), and vessel size. The bottom row displays hypoxia-inducible factor 1-alpha (HIF-1α) expression, vessel density as determined by CD31 expression, and vessel diameter based on CD31-highlighted vessels. These measurements are derived from targeted biopsies situated within three VOIs: nonenhancing, enhancing, and necrosis VOIs. Statistically significant differences based on the Mann–Whitney U test are marked with *, with *p*-values detailed above each respective comparison.

**Figure 7 cancers-16-00138-f007:**
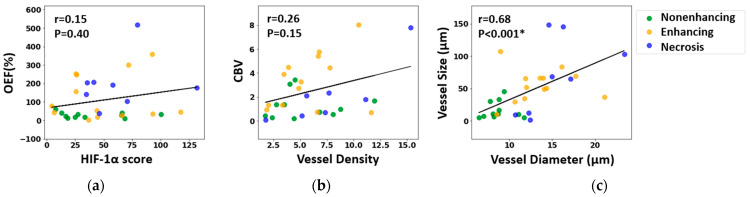
Correlation between MRI-derived metrics and histological characteristics: (**a**) oxygen extraction fraction (OEF) versus hypoxia-inducible factor 1-alpha (HIF-1α) expression; (**b**) cerebral blood volume (CBV) versus vessel density, determined via CD31 expression; (**c**) MRI-derived vessel size compared to vessel diameter measured from CD31-highlighted vessels. The strength of these correlations was assessed using the Spearman correlation test, and both the correlation coefficient (r) and the associated *p*-value (*p*) are reported, with * denoting statistical significance.

**Table 1 cancers-16-00138-t001:** Patient characteristics and the location of targeted biopsy samples. VOI: volume of interest.

Patient No./Sex/Age(y)	Pathologic Diagnosis	# Samples from Necrosis VOI	# Samples from Enhancing VOI	# Samples from Nonenhancing VOI
1/M/56	Oligodendroglioma (grade 2)	-	-	4
2/F/74	Astrocytoma (grade 2)	-	-	4
3/F/52	Brain Metastasis (lung carcinoma)	-	2	-
4/M/57	Brain Metastasis (adenocarcinoma)	2	1	-
5/M/72	Brain Metastasis (adenocarcinoma)	-	-	2
6/F/40	Oligodendroglioma (grade 3)	-	2	1
7/F/78	Brain Metastasis (adenocarcinoma)	-	3	-
8/M/75	Glioblastoma (grade 4)	3	1	-
9/F/68	Brain Metastasis (melanoma)	1	3	-
10/M/75	Glioblastoma (grade 4)	2	2	-
		Total: 8	Total: 14	Total: 11

**Table 2 cancers-16-00138-t002:** Summary of the quantitative analysis of the quantitative MRI metrics for various brain tissue VOIs across patients, represented as mean and standard deviation. VOI: volume of interest; contra-GM: contralateral gray matter; R_2_′: reversible transverse relaxation rate; DBV: deoxygenated blood volume; OEF: oxygen extraction fraction; CBV: cerebral blood volume.

VOI	R_2_′ (s^–1^)	DBV (%)	OEF (%)	CBV	Vessel Size (µm)
Contra-GM (*n* = 10)	5.13 (1.74)	4.30 (1.60)	43.89 (16.88)	1.66 (0.33)	16.24 (3.14)
Edema (*n* = 4)	3.21 (0.58)	7.43 (2.73)	21.06 (7.08)	0.71 (0.13)	16.78 (5.72)
Nonenhancing (*n* = 4)	2.56 (0.33)	7.48 (0.89)	14.90 (2.08)	1.04 (0.44)	12.97 (6.74)
Enhancing (*n* = 8)	7.90 (5.54)	6.44 (4.12)	60.81 (44.88)	1.78 (1.41)	25.73 (22.68)
Necrosis (*n* = 5)	7.36 (6.67)	6.72 (4.45)	54.15 (52.34)	0.70 (0.78)	16.61 (16.52)

## Data Availability

Data are available on request.

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
