# Peer review of "MRI-Based Assessment of Brain Tumor Hypoxia: Correlation with Histology"

_cancers, 2023, doi:10.3390/cancers16010138_

Round 1

Reviewer 1 Report

Comments and Suggestions for Authors

1- The resolution of some of the figures is not clear and should be improved.

2- All equations should be numbered and referred to the corresponding number in the text.

3- Please provide the number and demographic information of patients in the text.

4- The manuscript has no appropriate description of the results. All figures should be described and data should be compared statistically.

5- Figure legends should be provided self-explanatory in detail. Also, all abbreviations used in tables and figures should be defined.

6- Minor editing of English language grammar and spelling is required.

Additional comments:

The main question was addressed by the research, which was the relationship between MRI-derived measurements and histological data.

I consider the topic original and relevant in the field. It addresses a specific gap in the field by employing sqBOLD MRI to assess the OEF.

Compared with other published material, what the study adds to the subject area is the measurement of how much oxygen is being extracted from vessels. Also, utilizing Vessel Size Imaging (VSI) to evaluate microvascular dimensions and blood volume. This study employed streamlined-quantitative Blood Oxygen Level Dependent (sqBOLD) MRI to assess the Oxygen Extraction Fraction (OEF)—a measure of how much oxygen is being extracted from vessels, with higher OEF values indicating hypoxia.

Specific improvements the authors should consider regarding the methodology is there being no patient demographical information.

The conclusions are consistent with the evidence and arguments presented and they address the main question posed, including in the following sentence: 'The OEF, CBV, and vessel size maps provided insights into the tumor’s hypoxic condition, showing intertumoral and intratumoral heterogeneity.'

The references are appropriate.

Additional comments on the tables and figures:

Figure legends should be provided self-explanatory in detail.

Comments on the Quality of English Language

Minor editing of English language grammar and spelling is required.

Author Response

We thank the reviewer for their thorough review of our manuscript. Please find our point-by-point responses to the reviewer’ comments in the following document. 

Reviewer 2 Report

Comments and Suggestions for Authors

Comment: This is an interesting study and the authors have collected a unique dataset using cutting edge methodology. This paper has good conception and validation in crucial decisions for clinical brain tumor management.  The paper is generally clear, and structured. Sufficient information about this study findings is presented for readers to follow the present study rationale and procedures. However, in my opinion the paper has some imperfections in regard to some data analyses and text, and this unique dataset has not been availed to its full extent. 

Below I have provided considerable remarks on the text. In several instances I also suggested to cite more relevant literature. Furthermore, I made additional suggestions for more in-depth analyses of the data. 

Key critical points are: 

(1) The biggest issue is that the manifest data should be more robust and so, the authors are going to have to find a way to demonstrate their solid data even with only the ten subjects. 

(2) Figure 2: The VOI-averaged FLAIR-ASE signal across various VOIs for five patients with brain tumors. What is the important role and impact in this  image feature for results? It should be mentioned in the discussion. Furthermore, what's imply it and correlate with hypoxia in this topic research?

(3) The methods are generally appropriate, providing a rationale for the use of medical physics of measuring in hypoxia and correlation with histology . Scientific evidence is useful in clinical application. However, the sample size is too small, furthermore, how to validate more scientific merits and clinical implications in the research.

(4) All statistical methods should be individually shown in Materials and Methods.

(5) Histology is an attempt to strengthen consolidated correlation in this research even with only the ten subjects. Figure 5. resolution was blurred. Additionally, it should be  shown the magnification in the legend.

(6)This section emphasizes cancer informatics and big data, even though this research is start from image processing and analysis, it still has several points need to conquer such as imaging biomarker application in quantitative/qualitative imaging-derived metrics, and its development has shown the maturation of these technologies by demonstrating their clinical value. It seems the authors are writing about MRI physics as the authors explain some points about fundamental medical physics. This may be arduous for readers in cancer informatics and big data in this section. Some of the Radiology journals prefer multidisciplinary research and clinical applications in this novelty article such as Magnetic Resonance in Medicine, Journal of Magnetic Resonance Imaging, etc.. If the authors have more choice to try it.

Comments on the Quality of English Language

None.

Author Response

We thank the reviewer for their thorough review of our manuscript. Please find our point-by-point responses to the reviewer’s comments in the following document.

Round 2

Reviewer 2 Report

Comments and Suggestions for Authors

(4) All statistical methods should be individually shown in Materials and Methods.
The authors should add in 2.9 Statistical Analysis in Materials and Methods.

Author Response

Dear Reviewer, 

I apologize for any misunderstanding caused by the initial presentation of the statistical analysis. I have now added a separate section in our manuscript to comprehensively cover the Statistical Analysis. 

2.8. MRI measurements in the target voxels

To evaluate the correlation between MRI and histological findings, values for the MRI-derived parameters (OEF, CBV, and vessel size) were obtained by averaging these metrics from each target voxel from which biopsies were taken, and their adjacent nine voxels in three dimensions in postcontrast T1W images. This methodology was implemented to mitigate potential inaccuracies stemming from relying solely on single voxel evaluations and to accommodate for potential discrepancies caused by intraoperative brain shifts and MRI data registration challenges.

2.9 Statistical analysis

All statistical analyses were conducted using SciPy (version 1.10.1) software in Python (version 3.6). First, the Kruskal-Wallis H test, a non-parametric statistical test, was employed to evaluate the differences in various MRI biomarkers and histology parameters across three distinct regions of interest: Nonenhancing, Enhancing, and Necrosis. Post-hoc pairwise comparisons were conducted using the Mann-Whitney U test to further investigate specific differences between these regions and the results were annotated with their corresponding p-values directly on the bar plots.

Furthermore, correlations between MRI-derived measurements and histological counterparts were evaluated using a non-parametric Spearman correlation test. Examined correlations included OEF vs. HIF-1α score, CBV vs. vessel density, and vessel size vs. vessel diameter. For each analysis, the correlation coefficient (r) and its associated p-value were recorded, with a significance threshold set at P < 0.05. 

Thanks.
